# Systematic Review of Education for Sustainable Development at an Early Stage: Cornerstones and Pedagogical Approaches for Teacher Professional Development

**Martín Bascopé [1],\*, Paolo Perasso [2] and Kristina Reiss [3]**

[1]   Campus Villarrica, Pontificia Universidad Católica de Chile, Villarrica 4930445, Chile
[2]   Centro de Desarrollo Local, Educación e Interculturalidad (CEDEL UC),
     Pontificia Universidad Católica de Chile, Villarrica 4930445, Chile; paoloperasso@gmail.com
[3]   School of Education, Technical University of Munich, 80333 Munich, Germany; kristina.reiss@tum.de
\*   Correspondence: mbascope@uc.cl

**Abstract:** During recent decades, education for sustainable development (ESD) has been under the scope of the international community, but research in ESD for early childhood education (ECE) is still scarce. This article proposes a procedural framework for the implementation of teacher professional development opportunities in the area. During the first phase, we undertook a systematic review of peer-reviewed articles on ESD for ECE ($n$ = 30). After an expert committee revision of the articles reviewed, three cornerstones (scientific action-integrated, community-based and value-oriented scopes) and three sets of suitable pedagogical approaches (art-based, outdoor-based and project-problem-based) were identified. The review was enhanced by an unsystematic review of articles ($n$ = 26) that specifically referred to the cornerstones and approaches. Finally, a double-blind expert coding and categorization of the articles ($n$ = 56) was performed in order to validate the results. Focusing on guidelines and approaches, different examples found in the literature are presented. This review offers a useful framework to understand and practice ESD in ECE. Unlike previous reviews, it has a practical scope to foster innovative teacher professional development opportunities, inspire teachers and inform policy makers. We conclude with some common challenges and the needs for educational systems to foster science-based citizenship education towards sustainable development in a practical way, fostering agency from an early stage to transform local context, creating global awareness of the environmental, social and economic challenges of the 21st century.

**Keywords:** education for sustainable development; early childhood education; project-based learning; outdoor education; art-based learning

## 1. Introduction

Sustainability is a recent concept. The term first entered the *Oxford English Dictionary* in 1965. In the 1970s, there were only a few books with titles that contained the word, 'sustainability.' In the last decade, the titles of more than 10,000 books contained the word. Now, a search for the word 'sustainability' using Google will yield more than 150 million results [1].

Recent claims by climate scientists regarding global warming and climate change resonate with a sense of the need for changes in the educational system to tackle these global problems. Hamilton (2010), along with the Intergovernmental Panel on Climate Change (IPCC) reports [2], stresses the need to act now to confront and mitigate the consequences of climate change. Anderson and Bows [3] state that the 2 °C average increase in the global temperature is classified as extremely dangerous and

that the chances to reduce this increase are very low. Moreover, international meetings between policy makers have consistently failed in creating a consensus on the common goals needed for the reduction of carbon emissions [4].

During the decade between 2005 and 2014, UNESCO staged a worldwide effort to foster education for sustainable development. The United Nations (UN) report showed the extent to which policies have changed in that direction. Most stakeholders were convinced that solid progress had been made in all countries [5]. Still, the UN reported little empirical evidence of relevant change regarding pedagogical approaches, especially in early childhood education (ECE).

The UN has made a statement on the main goals and targets for 2030. The main content and ability-based target regarding educational systems (the 4.7 target) summarizes the complexity of the challenge and the urgency to translate it into proper professional development opportunities:

> "By 2030, ensure that all learners acquire the knowledge and skills needed to promote sustainable development, including, among others, through education for sustainable development and sustainable lifestyles, human rights, gender equality, promotion of a culture of peace and non-violence, global citizenship and appreciation of cultural diversity and of culture's contribution to sustainable development." (United Nations, 2018, Sustainable development Goals, 4. Quality education goal, target 7)

This places great emphasis on a wide range of themes and the need to build capacities from an early stage. More than a decade ago, the UNESCO Chair for Sustainable Development called for research into, and the practice of, education for sustainable development (ESD). However, it is still difficult to address this challenge in its full complexity.

The context-based focus that must characterize ESD applications should encourage the development of pedagogical frameworks, approaches and practical tools to be applied at a wider scale. Applications of ESD must be context driven, time dependent and open ended [6]. Every locality faces particular challenges in terms of human-environment and human–human relationships. Therefore, researchers and practitioners should work on specific ways to deal with them. We argue that, beyond specific contents and emphasis, there are certain common agreements in the literature about the profile we trying to raise with early childhood education. We believe that considerations such as these could serve as guidelines for more-than-local implementation. Based on the literature, we argue that certain pedagogical tools are key to professional development in ESD for ECE. Against a view of young children as not ready to be taught about critical societal issues, the literature tends to agree not only about the pertinence of ESD in early-childhood, but also on the necessity to foster child-centered approaches to start the development of abilities, attitudes and values for sustainable development from an earlier age.

From environmental education (EE) to ESD, environmental concerns are not seen as separate from social and economic issues. ESD requires a move from linear and analytical concepts to more systemic and complex concepts [7–10]. As was established by Biasuti [11] the three UNESCO pillars for sustainable development must be present (economy, society, environment), interacting with a fourth educational dimension to put sustainability into educational practice. Therefore, sustainability should not be considered as the sole "environmentality" anymore, particularly if the point of ESD is to educate for action. This article contains a science-based interdisciplinary framework for ESD for ECE which defies the traditional teaching and learning methods.

Research on ESD for ECE has experienced growth after the call by Davis et al. [12] to deepen research. However, further research is still needed. There are relevant case studies, as well as important efforts, to systematize the current research. Compared to ESD for secondary and higher education, ESD for ECE has been less addressed in scientific publications. In order to develop pedagogical tools that could be used to train pre-service and in-service teachers in ESD, it is important to understand how ESD in early-childhood has been practiced.

The complex problem of sustainability implies a big challenge for teachers and educators to implement strategies in a comprehensive and integrated way. As has been pointed out, the pedagogical approach on ESD for ECE is central [13] and there is a need for further research on particular pedagogical strategies to develop key ESD dimensions [14,15]. This paper presents an evidence-based framework for the practical application of educational activities for sustainable development in early-childhood education. After a systematic review of literature (*n* = 30), followed by an expert review of an international committee of practitioners, three cornerstones for ESD (scientific action-integrated, community-based and value-oriented scopes) on early childhood, and three sets of pedagogical approaches (art-based, outdoor-based and project-problem-based) were identified. Further literature that specifically addressed those cornerstones and approaches was added to the general review. Finally, two blind expert reviews checked the inclusion of the six categories and a coding process of the reviewed articles was performed in order to validate the outcomes of the review.

## 2. Research Question and Goals

Research question: How has research defined and described the implementation of education for sustainable development in early childhood education?
Objectives:

1. Describe and systematize an emerging and complex research field, from a trans-disciplinary perspective.
2. Understand the main guidelines for ESD for ECE and propose a research-based framework.
3. Classify the main pedagogical approaches present in the literature for the implementation of ESD for ECE.
4. Propose practical guidelines for the implementation of teacher professional development on ESD for ECE.
5. Inspire teachers to find proper opportunities for professional development on ESD.
6. Inform policy makers about the new trends of ESD for ECE

## 3. Context

In the current Latin American context, historical challenges in education, as means of social justice and interculturality, are fueled by climate change challenges and a growing crisis of democratic systems. Two of our authors live and work in the Araucanía region of South Chile. Here, environmental problems, such as water scarcity and deforestation, are linked to historical processes of land use, which are grounded in ethnic conflicts. Therefore, environmental and intercultural issues are present and interlinked. The region is well-known for its natural settings and series of national parks. It has also gained recognition as the homeland of the Mapuche people, who are engaged in a political struggle. The University Campus in Villarrica (Pontificia Universidad Católica de Chile, Campus Villarrica) specializes in education, granting undergraduate degrees in early-childhood education and primary education. It plays a key role in training teachers for the southern regions of the country, which are characterized by high rates of poverty, as well as indigenous and non-indigenous rural populations. Research on interculturality and local development is being carried out by two research centers, the Local Development Center (CEDEL) and the Center for Indigenous and Intercultural Research (CIIR). Because of the work of these two centers, ESD has become a concern for the institution.

Since 2014, Campus Villarrica, with the support of Siemens Stiftung, has developed relevant education programs aiming at the so-called STEM subjects, namely science, technology, engineering, and mathematics. These subjects are of specific importance for the 21st century as they are, for example, the basis to understand and evaluate phenomena such as climate change or consumption of natural resources. However, the programs mentioned do not aim at the transmission of declarative knowledge in a specific subject but support teachers in using problem-oriented as well as project-based approaches (PPBL) in their classrooms. Both aspects are combined in order to ensure a strong focus on inquiry-based learning of their students. Moreover, the programs address intercultural

and environmental education in the STEM classroom. After training more than 300 teachers, we are going to assess the impact of their education on their students' skills, abilities, and values. In parallel, we want to deepen the impact of our training by developing a set of pedagogical tools adapted that support the development of ESD to teach children from pre-school up to the 4th grade. Therefore, we plan to develop a set of relevant pedagogical tools to be used as guidelines to build indicators to assess previous and future training.

This learning process has been shared with and fueled by an international network of trainers under the coordination of the German foundation "Haus der kleinen Forscher" (Little scientists' house). A group of international experts met in October 2018 to plan an International Dialogue on STEM (IDoS), with a specific focus on ESD. The meeting provided participants with the chance to discuss pedagogical approaches by which they could promote local adaptations and foster action that would advance the educational systems towards sustainability. This group of experts, from 10 different countries across the five continents, discussed a common framework for ESD. Part of this work is expressed in this article.

This study presents the process by which we hope to re-orient professional development opportunities and to develop local application of ESD particularly in the south of Chile. The collaboration of this specific program with an international network concerned with ESD was crucial to validate the perspectives about sustainability in a global way. The framework focuses on the problem of how to promote local actions that would lead learners to think globally from an early age. If our framework succeeds in implementing local adaptations, it could foster similar initiatives elsewhere.

## 4. Methods

The methodology followed to reach our main findings can be summarized in Figure 1.

### First literature review
- Focus on Early Childhood Education
- Searching for scope and approaches for ESD

### Expert review
- Definition of emerging categories
- Comparision with other international experiences

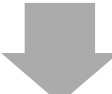

### Categories confirmation
- Double-blind expert categorization of the articles
- Coding and content analysis to check the categories.

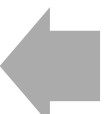

### Second literature review
- Non-systematic review, inspired by emerging categories.
- Focus on cornerstones and approaches discovered during the first review

**Figure 1.** Four phases of the methodology for the systematic review. ESD: education for sustainable development.

The review started by focusing on peer-reviewed publications only, specifically about ESD for ECE. We discarded publications about actors' perceptions on ESD or sustainability, as well as policy-analysis on ESD for ECE. For each publication, we analyzed: (1) What contents need to be addressed? (2) Which

abilities, attitudes and values need to be developed? (3) Which pedagogical approaches are being proposed? Some publications were reviews or discussions that addressed controversies over ESD, discussing appropriate contents, values or pedagogical approaches.

This first phase search was conducted in three scientific databases of peer-reviewed articles: Web of Knowledge [16], SCOPUS [17] and SciELO [18]. The keywords for the search were ("Early Childhood Education"+"Sustainability") and ("Early Childhood Education"+"Education for Sustainable Development") in both English and Spanish. Table 1 summarizes the criteria for the review.

**Table 1.** Criterion for the selection of reviewed articles.

| Criterion | Inclusion | Exclusion |
| --- | --- | --- |
| Type of documents | Peer-reviewed articles and Reviews | Books, chapters, conference papers. |
| Databases | Web of Science; SCOPUS; SCIELO | Other databases |
| Language | English and Spanish | Other languages |
| Sustainability focus | Education for Sustainable Development as a central part of the argument | Sustainability as a result of another researched educational aspect |
| Educational level | Early childhood | Primary, Secondary and Post-secondary education |
| Focus of the analysis | Methodologies and guidelines for the practice of ESD | Policy analysis, belief studies, curricular studies |

The first result after searching in the three databases was 159 peer-reviewed articles. After eliminating duplicates, we screened the abstract of 97 articles using our criteria expressed in Table 1. After this screening, 30 articles were selected for full text assessment, dating from 2003 to 2019, all published in journals of education, mostly environmental, early childhood and science education.

During the second stage, we presented a draft report with the main contents, skills, attitudes and values related to ESD at the International Dialogue on STEM 2019 committee meeting held in Berlin in October, 2018, in order to validate the preliminary results of our literature review ($n = 30$). Due to the presence of practitioners from several countries, the insights of the committee were invaluable in strengthening our argument and making our research more pertinent. Since ESD is a broad concept which involves a number of contents, skills, attitudes, and values, we proposed three cornerstones, or critical nodes, for ESD for ECE, as well as three main pedagogical approaches appropriate for this educational profile.

It is important to clarify that, in addition to ECE and ESD, we were interested in articles with specific approaches for ESD or guidelines for practice. Our goal was to ensure that this review would be a concrete contribution to the field, one that would inspire and generate more professional development opportunities for teachers. That is why our third step was to search for peer-reviewed publications addressing the specific pedagogical approaches and the ESD cornerstones ($n = 26$). We looked for studies that addressed the approaches found in the first stage. This review was non-systematic since it had the objective of deepening the emergent categories by presenting some examples from the field of educational research. We applied a snowball method with references found in the literature and the criteria used to end the searching was to reach a saturation point, when no new information arose from the articles screened.

To clarify the documentation reviewed in phases 1 and 3, a flow chart of the journal articles reviewed, following the PRISMA 2009 guidelines, is presented in Figure 2.

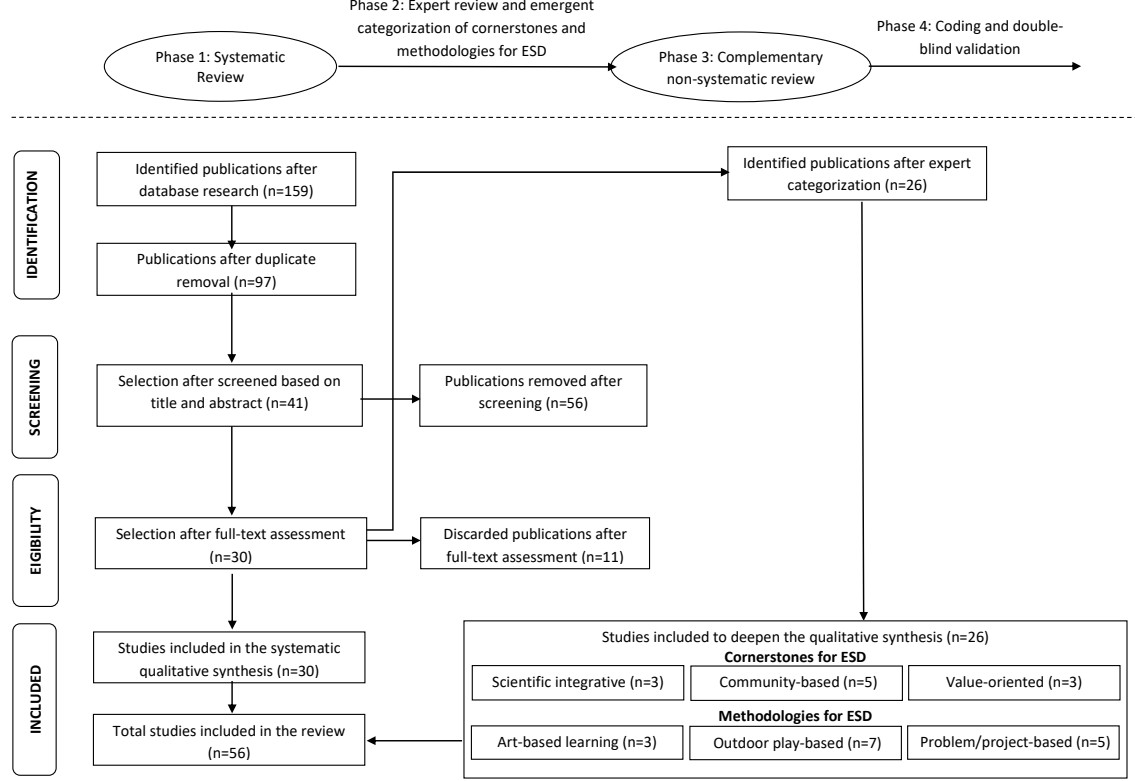

**Figure 2.** Number of publications reviewed according to the phases of the methodology, following the PRISMA flow diagram.

We refrained from searching systematically for literature that did not mention explicitly "Education for Sustainable Development" or "sustainability" as a key concept. No educational literature on related themes (e.g. intercultural education, environmental education, place-based education, etc.) was included in this review, if its relationship with sustainability was not explicit.

The fourth stage of the methodology was to confirm the categories proposed (cornerstones and approaches). We organized two blind expert reviews to check the inclusion of the six categories in each of the 56 articles reviewed. Using "Atlas.ti" software, we performed a coding process of textual quotes to confirm the expert analysis and to establish a definitive categorization of the articles when there was no agreement among the criteria. The coding process consisted of the extraction of textual quotes from the 56 articles reviewed, using a set of predefined codes based on different dimensions of each cornerstone and pedagogical approach issued from the first and second literature review plus the expert review. However, emergent codes were included, helping to enrich the description of each category. The results of the double-blind review reached 76.8% coincidence between experts' criteria to classify each article in the first round. After the coding process, the differences were clarified, reaching a final classification agreed upon by the experts. In this sense, we present an exploratory study to articulate guidelines, procedures and pedagogical approaches to implement professional development programs of ESD for ECE, based on the international experience collected in the scientific literature, supervised by an international expert committee of practitioners and confirmed with a double systematic qualitative process of classification and coding.

## 5. Results

ESD represents a theoretical effort in education to move towards the integration of fields of knowledge, to develop an inter-transdisciplinary approach addressing sustainability [8,19] in order to educate children to become change-agents grounded in places, and to advance constant reflection on pedagogical approaches and cultural assumptions guiding sustainability. On ECE,

ESD addresses the flaws and limitations of current early-childhood education and sustainability approaches. This encompasses the need to overcome adult-centric views of children that consider them incapable of coping with harsh and complex issues (e.g., climate change) [12,13,20–22], to focus on developing experience rather than transmitting knowledge to raise change-agents [13,23,24], to overcome positivistic and romanticized views in our relationship with the environment [20,25–28], and to integrate individual and collective values [13,29]. ESD encompasses previous fields of education, such as environmental education, science education, intercultural education, citizenship education and community-based education. It attempts to integrate them by means of child-centered approaches, in order to build a consistent framework to address the current socio-environmental crisis.

The literature on ECE is much more focused on skills, attitudes, and values than on specific content knowledge. Even though there is more research in the environmental content area, several studies focus on the social science side (indigenous knowledge, interculturality, and community-based knowledge). Other studies promote inter- or trans-disciplinary approaches. To summarize, students need to be able to cope with complex situations and to solve problems critically through collaboration with others [19,30]. Students should learn to think and act autonomously, and to become agents of change within the context of their own lives [12,29,31]. Therefore, the values framework needs to move from individual to collective values, without losing anything from aesthetics to ethics on the way. In other words, the values framework must foster and defend democratic citizenship values (respect for diversity, value democratic participation, respect the opinion of others, etc.) [8,32], while acknowledging and promoting collective values (respect for indigenous rights, value of the common goods, and respect for non-human-others) [20,26,33] and aesthetic values (to value cultural and environmental heritage at local and global levels).

Even though the research emphasizes the role that students and local communities need to play in the educational process, research on ESD, itself, has been questioned for not providing much space for children's participation as agents for change [23,25,34]. In this sense, it is important to not only be aware of the main contents, abilities and values related to ESD, but also to know how to implement it and what needs to be present in any ESD project. The following section presents a proposal on how to implement ESD for ECE, with a special focus on defining the characteristics and procedures that the evidence presents for implementation. We do not use the typical scheme of knowledge, skills, attitudes, and values related to a specific topic. The field of ESD is too wide to make a useful list in this respect. Instead, we have gone one-step further and described three main cornerstones and the adequate procedures to develop knowledge, skills, attitudes, and values towards sustainability, to build a useful and consistent framework for teacher professional development.

### 5.1. Cornerstones for the Implementation of ESD Activities

Looking through the literature on ESD, we highlight certain common critical points needed to orient the pedagogical approaches. These critical points must be understood as not intrinsically being part of the approaches, since those approaches are not exclusive to ESD. However, the use of those approaches in an ESD context is made possible through the acknowledgement of those critical nodes. We are going to expand on the three cornerstones for ESD activities: (1) they should be action-integrative, meaning they should encourage practical activities that integrate scientific and non-scientific knowledge; (2) they should be community-based, and encourage relevant activities with transformative goals; and (3) they should be value-oriented, with a large enough scope to develop ethics and aesthetics attitudes regarding humans and non-humans.

Based on the experts' categorization and the coding process, the following sections will describe in more depth what was found in the literature. The three cornerstones were present in almost half of the articles reviewed. The scientific action-integrative scope was present in 45.1% of the articles, the community-based cornerstone in 49.0%, and the value-oriented focus in 51.0% of the literature reviewed. Hence, their presence was balanced in the literature reviewed and many of the papers tackled more than one of the cornerstones.

### 5.1.1. Scientific Action-Integrative Scope

The global environmental situation demands an education that will promote agency starting as early as possible. The development of agency is considered of major importance for ESD [12,29,35]; however, as research shows, it has not been put into practice [8,31]. Therefore, ESD pedagogical approaches need to concentrate on raising change-agents to impact their communities through an action-oriented, participatory and integrative focus [36]. It needs to be an experiential learning approach because knowledge content transmission is not enough to raise agents, as Frisk and Larsson [9] argue. Environmental education based mainly on transmission of declarative knowledge has not led to behavioral change on sustainability issues. Referring to ECE, content transmission is not what early-childhood education is about [13,23] but rather it should focus on developing experiential learning among children [37]. In order to raise change-agents with ESD, particularly from early-childhood, education needs to focus on promoting meaningful experiences for developing agency [29]. Therefore, agency should be considered as an "engagement with particular temporal-contexts for action" rather that something to be possessed [25] (p. 441), which reminds us of the importance of learning environments focused on developing experience [12,27]. For example, recommendations made in the 2009 special review of ESD for ECE "Early Childhood Education for Sustainability: Recommendations for Development", state that children, teachers, and all actors in the educational community need to practice sustainability (e.g., through recycling, gardening, energy production) daily and not just through teaching [38]. It is important to be cautious in fostering agency. Care should be taken to avoid putting on children's shoulders the sole responsibility for the environmental damage caused by previous generations [38].

To create environments that foster agency through action-based learning, education must integrate fields of knowledge and experience. Children need to be considered active stakeholders in sustainability issues [22,29,35] and to be encouraged to become problem-seekers and solvers in their own localities [12]. This relates to the relevance of scientific literacy among young children. They need to learn to undertake science, to acquire scientific competences through practice, which presupposes the integration of fields of knowledge (interdisciplinarity) to address practical issues. Furthermore, as we will show in the next section, ESD needs to be based on the active involvement of the local community, which calls for integrating science with other forms of knowledge present in the locality (i.e., trans-disciplinary).

### 5.1.2. Community-Based

Different authors agree that ESD, starting from ECE, needs to involve and impact local communities [7,12,39,40]. The degree and kind of involvement with communities may vary according to the context. It may start with regarding local communities as a learning field (local, indigenous, etc.) [39], and may lead to situations where children invite other members of the community to participate in a process of co-learning with students, oriented toward solving local sustainability challenges. As Tilbury and Wortnam [41] state, the community-based orientation of ESD is grounded in the relevance of non-formal community education.

The term "community" can be understood differently according to the context. In general, we noticed that authors do not clearly define it. Usually it refers to the educational community (parents, neighbors, school workers, etc.) but it can also refer to the locality at large. In the indigenous context, it usually refers to a form of political organization dwelling in the territory where the school is located. However, what defines the community-based orientation of ESD is the idea of education as a localized social process encompassing all actors directly or indirectly related to the school in order to address sustainability challenges through an action-oriented approach. Different strategies were applied to develop community-based ESD projects. We found three forms of community involvement:

*Learning from and about the community*: Children learn from their own communities through different forms of participation in community life (social, political, cultural, etc.) and engaging in dialogues with community actors. It also allows diverse forms of knowledge to be part of the pedagogical process, meaning that the knowledge present in the community should find a place in the educational process

and, hopefully, be taught on an equal footing with scientific knowledge [20,22,42–44]. As an important component of what Duhn [22] calls "pedagogies of place", this contributes to developing a "sense of place" by inquiring about local knowledge and heritage and contributing to local identity development [39], which is central in coping with global challenges.

*Acting upon community issues*: Children are agents in the transformation of their own localities. Therefore, they develop strategies to solve local sustainability challenges with adult guidance. The objective is to raise change-agents actively involved in sustainability measures in their own locality [29,34].

*Co-learning and acting with the community*: Community actors (particularly parents) collaborate in the educational process and learn with children through action-oriented projects to solve local sustainability problems [40,41]. These forms of community participation help to build capacities among families and communities [7,12] in order to collectively face sustainability challenges. Therefore, the objective for local schools is to become sustainability centers, which will have an impact at the local level.

The three forms of involvement can be thought of separately because they target different pedagogical and societal goals. They are not exclusive but complementary. In general, we found that at least two of three were present in most community-based ESD interventions.

### 5.1.3. Value-Oriented

Directly or indirectly, most of the literature has stressed the relationship between ESD application, and ethical and aesthetical formation. ESD challenges are thought to be partly an issue of values either because value formations represent the core of early-childhood education itself [13,32] or because global sustainability challenges reflect a civilization crisis which demands radical changes in how we relate to others (humans and non-humans) [7,8,20,26].

We must differentiate among ethical and aesthetic values, although both are of fundamental importance for children's development. ESD needs to develop values that promote a democratic citizenship [23]. As Green [34] points, the UN Convention on the Rights of the Child considers children citizens who possess the right of participation. In addition, children need to be able to cope with diversity and to value it (in terms of gender, class, ethnicity, religion, etc.) in a globalized world [13,29]. Notions of equity and social justice are mobilized and can be strengthened in the daily interactions between children [13].

There is a claimed necessity, present in the literature, to develop values that have not been promoted in modern Western societies, and in the current educational system [27,30,43] as collective values. Children need to be able to cooperate with others. They need to protect the commons (as water, soil, forest, and local culture) which cannot be done individually but only through collaboration. Therefore, individual and collective values need to be developed together. Following Hägglund and Samuelsson [13], this relates to the recognition of the importance of ESD to educate the child to act independently while acknowledging dependency in relation to others.

Many authors working on intercultural contexts stress that value change in education needs to go beyond humanism and start to develop another way to relate with non-human beings. An ethic to care for other forms of life and for "place" must be promoted [20,27], realigning humans as part of "a multi-relational exchange of belonging with nature" [26] (p. 60) and not separate (i.e., either outside or over). Teachers can over-influence children's experiences by transmitting their own ideas and emotions regarding their personal and cultural relationship with the environment. Therefore, teachers should be aware of the pervasive influence of romantic and utilitarian views on the environment as part of a Western ontology dividing nature and culture [20,26,27]. It implies a way of relating with the environment where nature appears to be "out there", separated from "us" as human beings, leading many to think that humans could master it. On the other side, it nurtures romantic views of the relationship between children and nature (as the word kindergarten suggests), which sets aside critical engagement with implications for educating for sustainability. Values and worldviews based

on interdependence between humans and non-humans are important among indigenous peoples, which tell us about the potential of intercultural education for the transformation of values [43,44].

The forging of aesthetic values through direct and meaningful experiences is key to engaging children with a natural and cultural context. Art and play can bring about positive experiences, thereby developing an appreciation of the cultural and natural commons, as well as an optimistic attitude towards life. Aesthetics as a meta-structure helps to develop connections with distant objects, activating the learning process through intuition and creativity [45] and fostering innovation [6]. Therefore, developing aesthetic values has the impact on the cognitive development necessary to forge a sense of belonging with the local environment.

## 5.2. Suitable Pedagogical Approaches for ESD

Considering these three cornerstones for ESD for ECE, it is important to give some examples and practical approaches for implementation. The three pedagogical approaches presented in this section can integrate the three cornerstones presented and can be very useful for teachers and teacher trainers to make explicit the goals behind an ESD activity.

### 5.2.1. Art-Based Inquiry Experiences

Art is a booster of creativity and complex thinking, as well as the aesthetic evaluation of our environment. Through art, a sense of place and belonging can be developed by promoting an affective engagement with our surroundings. Art can incorporate meaning with scientific inquiry, environmental action and community place making.

As Eernstman and Wals [6] emphasize, art allows locative meaning within the context of sustainable development (SD). Against an over-intellectualized approach in the quest for a fixed and universal definition of the term, the authors stress that:

> ESD essentially starts with and revolves around re-embedding SD in life and the act of living. Instead of depending on scientific and abstracted descriptions of what SD should mean to people, learning for SD lies in processes that incite communities to yield their own context and time specific interpretations of sustainable development. [6] (p. 1657–1658)

Therefore, art is a way to ground SD in places, allowing people to make sense of it according to their own lived experiences. Reflecting its potential for transcultural understandings of ESD, art has also been used as a tool for communication and sharing views about the environment between children of different cultural backgrounds [46]. Even though art-based approaches are relatively rare in the literature (incorporated in only 14% of the articles reviewed), it is an innovative and meaningful way to tackle ESD for ECE. In addition, research results on how to foster meaningful experiences to incorporate abstract concepts by means of creative activities are very powerful. Hence, we decided to incorporate art as part of a pedagogical approach for its potential both in practice and research on ESD.

Art-based approaches are considered an important feature of ESD applications specifically for ECE. Illeris [47], for instance, advocates AESD (art + ESD) to foster environmental care by helping to break cultural assumptions that hinder action, being "a driver of critical and reflective thinking" [47] (p. 91). Therefore, it can contribute to the understanding of harsh and complex sustainability issues among young children.

On the other hand, Ward [48] uses creative art-based pedagogies to support children's engagement with their environments. Through relevant creative experiences, sustainability content is incorporated and expressed in meaningful ways. Through painting, acting, playing music and writing, children learn about local flora and fauna, and weather phenomena.

Sorin [46] shows how the use of art-based digital technologies can become useful for young children communicating beyond local contexts. Children from Canada and Australia shared their views and deep emotions about the environment online through postcards and storytelling. This intervention

led to an understanding of how children represent their environments and about the potential of art for ESD.

Chan, Choy, and Lee [49] show how the use of artistic activities and art crafts have a double meaning for ESD. With the support of artists-in-residence, they taught children about recycling while using waste as materials and taught about the human-environment relationship using symbols (e.g., the dolphin as a symbol of harmony). In both cases, art in ESD for ECE is used to transfer content on relevant sustainability issues.

According to Eernstman and Wals [6], art can also be used to connect school and community around ESD issues. Even if not referring specifically to ECE, they show that artistic collaborations between actors can have an impact on the collective memory and its uses in community-based ESD projects. Considering the literature on art and ESD, in early-childhood education there are paradigmatic examples of how art can become an inherent part of early-childhood education [45]. Spaces of artistic experimentation led by artists and teachers offer a great opportunity for children to develop a sensory engagement with the world. The use of diverse tools and materials helps children to freely express themselves and collaborate with others, thereby developing an aesthetic dimension that is grounded in strong ethical and political values [45].

### 5.2.2. Outdoor Education as a Basis for ESD Learning

When researchers refer to outdoor education, they differentiate between a range of pedagogical experiences, moving from adult-guided to free-based experiences aimed at developing a sensitive engagement with the environment and/or learning about it. In other words, it is not any outdoor activity, but those who deliberately aim at triggering long-lasting bonding between child and environment, as place (culturally signified local environment) and/or as "nature". Among those pedagogical experiences, some are framed as play, as outdoor exploration or inquiry, and as participation in community life.

Samuelsson and Johansson [50] have discussed the importance of play as learning and learning as play as part of the children's experience and process of creating meaning of the world. In this sense, play is a constitutive part of learning during early childhood years [31,37]. If framed according to a child's particular needs, play allows the development of cognitive functions, as well as communicative and social skills [51]. However, the characteristics of the playground and children–teacher interactions are a strong determinant of how children play and what they learn while playing [52]. Therefore, in ESD, outdoor play, purposely framed, is said to allow environmental knowledge formation [31,53].

Even if play is usually related to outdoor experiences in ECE, not all outdoor activities involve playing as a method. Green [33,54], for instance, argues for free exploration within natural settings in order to develop sensory engagements with place. To strengthen children's environmental exploration and agency, she proposed "Sensory Tours", a method to research children's experiences in nature and to allow them to freely explore the environment.

A range of studies [14,28,33,54], some of which directly question adult-centric education, advocate for outdoor experiences with minimal adult intervention. Defending children's autonomy and their need for adults to take them seriously as agents in their own environment [29], some research studies stand against "teacher-directed-methods" [14] as an appropriate approach to outdoor learning for ESD.

This view has been nuanced by others [53] who state that a certain content-orientation proposed by teachers is needed so that outdoor activities could have a greater impact. Outdoor experiences are important for developing sensory engagement with the environment, which is considered the basis for developing environmental stewardship [12,14,28]. From a critical and post-humanist perspective, Elliott and Young [26] have argued that, even if outdoor experiences are beneficial for children, they do not automatically produce an engagement with sustainability issues:

> " . . . one cannot expect the complexities of human–nature relationships and their implications for global sustainability to be addressed simply by humanist notions of stewardship and children playing in nature." [26] (p. 59)

Therefore, it is important to be reflective about our own cultural conditionings while teaching children about human-environment relations, particularly in an intercultural context where different ontologies may be at play [44].

Night and Bertely [44] applied the intercultural inductive methodology [55] in the indigenous context of Chiapas, México. By using the traditional agricultural system (milpa) as a starting point, most activities are outdoors but framed towards valuing the knowledge present in the community in order to integrate it symmetrically with the dominant scientific knowledge. Intergenerational dialogues have been widely applied in indigenous contexts for this purpose [20,44,56] and can be conducted in outdoor settings performing everyday activities. Similarly, in New Zealand, Ritchie [43] shows how the growing, consuming and sharing of food is informed by Maori worldviews and values. In both cases, outdoor experiences have practical outcomes as they contribute to community livelihoods, social bonding, and cultural enhancement in their own localities.

### 5.2.3. Project and Problem-Based Learning

Project and problem-based learning (PPBL) are two complementary pedagogical approaches widely used in ESD, STEM education and sustainability science. Both are oriented to action, integrating fields of knowledge (inter and transdisciplinary), and fostering the development of agency and collaborative skills among children. They engage students in real-world problems, considering them as active rather than passive learners who work to find solutions [30,57,58].

According to Brundiers and Wiek [57], problem-based learning engages students to inquire into problematic situations in a loosely predefined setting, helping to develop deeper understandings on particular issues. On the other hand, project-based learning engages students in project management to reach practical results, developing a case-specific understanding. The authors state that the combination of problem and project-based learning (PPBL) develops as follows: inquiry into a local problem or the involvement with local stakeholders is problem-based but the search for science –based solutions adopts a project-based focus. That is why we propose a PPBL approach, which contains both the relation with local contexts and the inquiry science-based scope.

As Wiek et al. [19] show, PPBL are useful in developing a series of competencies that, according to Bell [30], are urgently needed for the 21st century: systems-thinking competence; anticipatory competence; normative competence; strategic competence; and interpersonal competence. Frisk and Larsson [9] recognize PPBL as an experiential approach empowering children to be problem-solvers through hands-on inquiry and collaboration. They divide it into three components: a driving question that organizes a long-term investigation, the production of a tangible result, and the collaboration of diverse actors. Experiential learning approaches, such as PPBL, facilitate a community-based orientation of the pedagogical process, either as an inquiry about the community or as an action-oriented approach to solve local problems.

Therefore, the earlier we implement the PPBL approach, the better. ESD should start working with PPBL. There are important experiences of art integration into early-childhood education through project-based learning and child-centered approaches where children are encouraged to follow their own curiosity [45]. Therefore, we can observe that PPBL approaches are intrinsically action-integrative, encourage the development of collaborative skills affecting value formation, and have the enormous potential of facilitating community involvement in the pedagogical process.

## 6. Discussion: ESD as a Scientific-Based Framework towards Sustainable Citizenship

Our review aimed at understanding and specifying how education for sustainable development in early childhood education may be implemented. We used a transdisciplinary perspective and tried to systematically describe this emerging field of research. Unlike previous reviews, a major contribution of this work was the focus on how to put ESD into practice, what elements should be present, and what kind of pedagogical approaches could be used for this purpose. As a result, we suggested a procedural framework, rather than a theoretical one, based on international experience as

evidenced by the scientific literature. The framework is intended to provide information on activities and innovations that programs of education for sustainable development should encompass.

Nevertheless, some limitations of this study have to be considered. First, we have to acknowledge that even though the databases revised included a great number of peer-reviewed journals, neither articles outside these databases, nor books, nor book chapters, nor any other kind of publication were considered, which limits our results only to that universe. Second, the review disregarded all literature regarding teacher and student beliefs about SD or studies about learning communities and institutional capabilities to install ESD in educational institutions. A crucial step for implementing ESD in practice is related to beliefs, conceptions and institutional capabilities to implement and sustain an ESD initiative. More research in ECE about this topic is necessary as a complement of the present review, to contribute to the improvement of ESD practices.

In the scientific literature, there is a lack of profound descriptions or examples of the implemented pedagogical approaches at an early stage. Our contribution was to systematize and to give practical examples of what has been done. The next step for the program—particularly in Chile, but also within the international network on ESD—will be to publish well-systematized results from the implementation of these approaches, both in scientific journals and teacher publications. We also hope to disseminate the results by organizing congresses, seminars, workshops, and courses for teacher professional development.

Even though the local experience of professional development programs in Chile has proven that there are many teachers who are interested in ESD, the predominant focus on environmental issues and natural sciences has limited the impact of these professional development programs. The framework proposed here can encourage those responsible for teacher professional development to enhance the coverage and possible impact of their programs, with a wider scope and clear objectives. Moreover, it might serve as a guideline for teachers supporting the identification of proper courses for professional development. The framework can thus be used for identifying opportunities for teacher professional development, and for promoting policies on initial and continuous teacher training towards sustainability.

Historically, especially in Latin America, colonizing discourses have been very close to the scientific project, generating distrust in the educational systems. Asymmetries between scientific and non-conventional/traditional knowledge are an important issue to consider. We are not proposing integration but, to transcend the hierarchical discourses, to be open to create and to adapt new knowledge from an early age. In this direction, the ESD framework is a way to give a common orientation to the school systems, one with a transdisciplinary scope that considers indigenous knowledge, intergenerational exchange, scientific knowledge, traditional lifestyles, and local practices. In this respect, the framework might be a basis for policy makers in order to understand the importance of ESD for ECE and to support its implementation. As the framework is particularly based on a systematic review, there is empirical evidence for the cornerstones and pedagogical approaches suggested providing a sound fundament for political decisions.

*Final Remark*

In our view, there is a further consequence of our study: regarding the literature review and considering the diversity of scopes and methods to promote sustainability in ECE, we suggest as a conclusion that action is needed, and that acting to adequately share ideas and examples is an important issue. Hence, we propose the understanding of education for sustainability as part of citizenship education. The concept of citizenship can be a helpful way to understand the magnitude and complexity of the changes needed. Citizenship, as an interdisciplinary approach fostered by teachers from different backgrounds, encourages students' capacity to act, to think critically, and be transformative in their own contexts. It also empowers future generations to think and act differently, towards a better and more sustainable world. ESD must be understood as being transversal and

beyond disciplines; it goes more to the fundamentals of cosmopolitan citizenship and the way we interact with our context in our everyday life [23].

**Author Contributions:** Conceptualization: M.B., K.R. and P.P.; Methodology: M.B.; Validation, M.B. and P.P.; Formal Analysis, M.B., K.R. and P.P.; Investigation, M.B. and P.P.; Writing-Original Draft Preparation, M.B. and P.P.; Writing-Review & Editing, M.B., K.R. and P.P.; Supervision, K.R.; Project Administration, M.B.; Funding Acquisition, K.R.

**Funding:** This work was supported by the collaboration agreement stablished between Siemens Stiftung and Pontificia Universidad Católica, Campus Villarrica.This work was also supported by the German Research Foundation (DFG) and the Technical University of Munich within the Open Access Publishing Funding Programme.

**Conflicts of Interest:** The authors declare no conflict of interest.

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
