# Peer review of "Systematic Review of Education for Sustainable Development at an Early Stage: Cornerstones and Pedagogical Approaches for Teacher Professional Development"

_sustainability, doi:10.3390/su11030719_

Round 1

Reviewer 1 Report

It is altogether a very good article which is based on many articles

The procedure is quite clear, although I am not familiar with Atlas ti coding – but it look very relevant way of coding.

One main problem is that the notion of “method” is of concern, since it is not clear always if the authors talk about the method used in the various articles or if they really mean pedagogical or didactical approaches. Method gives a feeling of something very technical, and that is not what I think the authors talk about?

STEM should be explained.

The last analyses is for me the most questionable, is art-based inquiry experiences a method, and how did that appear to be so central, at the same time as the authors say that art-based methodologies (or pedagogies) are relatively rare in the articles? I also question the references to Reggio Emilia with Ateliers. There are excellent examples of Reggio preschool, but like all other branches (Montessori etc), they are not good because they call themselves a specific name?

 Under out-door education there is certainly a section about intergenerational dialogues? Why?

The section with “Project and problem-based learning is very” superficial, and once again Reggio Emilia is used, but there must be any references to projects or problem-based, which are two different approaches, that can be integrated, but do not have to be that.

I think the last section is what has to be improved, beside this is a very good article.

Author Response

Dear, 

Thanks a lot for your kind comments and recommendations, your input has been very useful to improve our manuscript. There is a lot of work behind it and we appreciate very much your interest. 

Please find attached our responses to your comments, indicating the row where the changes were made in the new version.  We hope that you our responses come up to your expectations

Best regards

The authors

Reviewer 2 Report

Manuscript: Systematic Review on Education for Sustainable Development at an Early Stage: Cornerstones and Educational Methodologies for Teacher Professional Development 

The review article is interesting to read and gives valuable information in the field of early childhood education/education for sustainable development. The area is not very well researched and this paper gives an overview of ECE. This article is worth to be published.

Abstract;

- includes mainly all important issues, however the aim of the study is missing – please add the aims of the study into the abstract

- the authors write in the abstract: ‘This article proposes a procedural framework for the implementation of teacher professional development opportunities in the area’. Obviously this is meant to cover the purpose of the study. Still this does not substitute the aims, which should be added in spite of what has been written of the purpose of the study. 

-please also add to the abstract the gap in research what this article is going to fill

Introduction

-background ideas for the authors how to clarify the purpose and the aims of the study given later in the text

-in rows 92-95 (at the end of the introduction) the authors write:’ Besides the aim to inspire teachers and teacher trainers to foster innovation towards ESD, the aims of this study are three. After this systematic review, we aim to give some guidelines for the creation of new professional development programs, informing policy makers of the new trends in the field on ECE and, of course, to help teachers and educators to look for proper professional development opportunities’. However, rather than presenting the aims in the introduction a separate chapter should be used (see below).      

-it is useful to write a separate chapter of research aims and research questions. The research questions would form the main thread through the aims, the results and the discussion. This chapter would structure the study for the reader. The research aims should be numbered (1, 2, 3) and followed by research questions, e.g. (research question 1) what kind of guidelines for the creation of new professional development programs in ECE could be given? One could also link the research questions to the scientific action-integrated, community-based and value-oriented scopes (three cornerstones) and to the art-based, outdoor-based and project-problem - based methodologies (three sets of suitable educational methodologies, which have been identified).

-“the term (sustainability) first entered the Oxford English Dictionary in 1965 “– do the authors mean in connection to sustainable environment? Please define the concept, also, what does it mean and especially in this study. In the title is sustainable development but here the word sustainability has been mentioned.

-please add also where this study lays theoreticallly (I understand that the goal is in practical matters, but general theoretical background help readers focus to the goal of the study)

-IPCC- and perhaps UN reports shortenings could be explained Note later in text United Nations has been written out …..(of course everybody should know them, but perhaps to harmony and the fluency understanding for reader) when mentioned first time

-Anderson y (??) Bows…. Anderson and Bows

-in row 35 policymakers but in row 94 policy makers – check and uniform the spelling

Methods

Figure 2 is inaccurate / blurred, please add the original figure 2

- in row 197: ”seven found but only one was suitable for the purpose of this study” ….Why not suitable?. 

-  which were the articles found? What were the Journals they had been published in ? What were the years they were published? What where the levels of the articles which have been analysed (scientific, international, peer-reviewed???)

- though it is not a question of the systematic review, however the methodology/analyses of this article are somehow difficult to combine with the results. Please clarify.

- procedure of Atlas ti -program is not explained such as what were the codes/words etc which have been used. Please explain.

Results

- it is written: ”Seven articles were found in the first identification phase and only one was suitable for the purpose of this study.” Please identify, what made it suitable in this case (the criteria have been introduced earlier in the text, please apply). This (seven articles)  was already written on page 197? Please check and remove here, this is not a result.  

- Please remove from the results the sentence “A large number of recent articles on ESD have been produced, with 61% of the total having been published since 2014.” Does this sentence mean the articles you have analysed? In that case you should give the number of the articles analysed and it should be written in the Methods. In case it is a general finding, what is the connection to your study? Maybe not at all and can be deleted?

- in rows 233-234 the authors write: We prefer to go one step ahead and describe three main cornerstones and the adequate procedures to develop knowledge, skills, attitudes and values towards sustainability. … please explain this result from the point of view of the study design not what do you prefer. 

- Is this information more likely part of the methods than the results?

- This approach seems to be right choice to present the information gathered in this study. It clarifies for the reader the entity.  

Please connect the following subtitles to the research questions:

Cornerstones for the implementation of ESD activities 

Scientific action-integrative scope

Community-based -in rows 295-303, check and correct the line spacing

Value-oriented -in rows 329, 330 the paragraphs starts ”But they need to develop values that have not been promoted in modern western societies, and in the current educational system [24,27,40] as collective values.” To what group do ”they” refer, please do not start a paragraph using but..

Suitable methodologies for ESD 

Art-based inquiry experiences 

Outdoor education as basis for ESD learning 

Project and problem-based learning 

Discussion: ESD as a scientific based framework towards sustainable citizenship 

- please structure the discussion according the research questions /aims, results (somehow these three parts stay quite separate in this version)

-please also refer to the articles which support your findings

I hope the authors find my recommendations useful and hope that these suggestions help the authors to prepare the manuscript to a more fluently readable form. 

All in all this manuscript gives useful information to readers.

Author Response

(The authors gave the same response as above.)
